# *LAMA2* Nonsense Variant in an Italian Greyhound with Congenital Muscular Dystrophy

**DOI:** 10.3390/genes12111823

**Published:** 2021-11-19

**Authors:** Matthias Christen, Victoria Indzhova, Ling T. Guo, Vidhya Jagannathan, Tosso Leeb, G. Diane Shelton, Josep Brocal

**Affiliations:** 1Institute of Genetics, Vetsuisse Faculty, University of Bern, 3001 Bern, Switzerland; matthias.christen@vetsuisse.unibe.ch (M.C.); vidhya.jagannathan@vetsuisse.unibe.ch (V.J.); 2Neurology-Neurosurgery Service, Willows Veterinary Centre and Referral Service, Solihull B90 4NH, West Midlands, UK; victoria.indjova@gmail.com; 3Department of Pathology, School of Medicine, University of California San Diego, La Jolla, CA 92093-0709, USA; liguo@health.ucsd.edu (L.T.G.); gshelton@health.ucsd.edu (G.D.S.); 4Department of Neurology and Neurosurgery, Anderson Moores Veterinary Specialists, Winchester SO21 2LL, Hampshire, UK; Brodeljos@hotmail.com

**Keywords:** *Canis lupus familiaris*, dog, muscle, neuromuscular disease, laminin, merosin, precision medicine, animal model

## Abstract

A 4-month-old, male Italian Greyhound with clinical signs of a neuromuscular disease was investigated. The affected dog presented with an abnormal short-strided gait, generalized muscle atrophy, and poor growth since 2-months of age. Serum biochemistry revealed a marked elevation in creatine kinase activity. Electrodiagnostic testing supported a myopathy. Histopathology of muscle biopsies confirmed a dystrophic phenotype with excessive variability in myofiber size, degenerating fibers, and endomysial fibrosis. A heritable form of congenital muscular dystrophy (CMD) was suspected, and a genetic analysis initiated. We sequenced the genome of the affected dog and compared the data to that of 795 control genomes. This search revealed a private homozygous nonsense variant in *LAMA2*, XM_022419950.1:c.3285G>A, predicted to truncate 65% of the open reading frame of the wild type laminin α2 protein, XP_022275658.1:p.(Trp1095*). Immunofluorescent staining performed on muscle cryosections from the affected dog confirmed the complete absence of laminin α2 in skeletal muscle. *LAMA2* loss of function variants were shown to cause severe laminin α2-related CMD in humans, mouse models, and in one previously described dog. Our data together with current knowledge on other species suggest the *LAMA2* nonsense variant as cause for the CMD phenotype in the investigated dog.

## 1. Introduction

Congenital muscular dystrophies (CMDs) are a heterogeneous group of inherited neuromuscular disorders. They show a wide variety of clinical features including early onset weakness and hypotonia, as well as delay or arrest of motor milestones and loss of those same milestones later in the course of the disease [1]. Serum creatine kinase (CK) activity is often elevated, suggesting disruption of the permeability of the sarcoplasmic membrane [2]. Muscle biopsies typically show dystrophic features [3]. To date, variants in at least 39 different candidate genes are recognized that lead to CMD in human medicine [1]. Among the most commonly observed CMDs are those resulting from variants in genes coding for structural proteins of the basal lamina and extracellular matrix, such as type VI collagen or laminin α2 [4]. Another CMD subtype that is seen with a similar frequency are the dystroglycanopathies, which arise from defects in the various genes involved in production and post-translational modification of α-dystroglycan [5].

The classification of CMD in veterinary medicine is comparable to that in human cases [6]. In recent years, several candidate causative variants for CMD subtypes in dogs were identified. Pathogenic variants in *COL6A1* in Landseer dogs (OMIA 001967-9615) [7] and in *COL6A3* in Labrador Retrievers (OMIA 002274-9615) [8] cause Ullrich type CMD, while a nonsense variant in *LARGE1* was associated with a dystroglycanopathy in a family of Labrador Retrievers (OMIA 002460-9615) [9]. Furthermore, a *LAMA2* variant was recently identified in an American Staffordshire Terrier with laminin α2-related muscular dystrophy (*LAMA2*-RD, OMIA 002459-9615) [10].

The current investigation was initiated upon presentation of an Italian Greyhound with a suspected congenital neuromuscular disease. The goal of the study was to characterize the clinical and histopathological phenotype of the dog and to investigate a possible underlying causative genetic defect.

## 2. Materials and Methods

### 2.1. Clinical Examination

A 4-month-old, male Italian Greyhound was investigated. Physical, orthopaedic and neurological examinations were performed. Clinicopathologic investigations performed included complete blood count, serum biochemistry including CK activity, electrolytes, *Toxoplasma gondii,* and *Neospora caninum* serologies. Electrodiagnostics were performed under general inhalational anesthesia. Electromyography was performed on right sided axial and appendicular muscles. Left-sided muscles and head muscles were not evaluated. Motor nerve conduction velocity (MNCV) was evaluated along the right tibial/sciatic nerve and left ulnar nerve. Repetitive nerve stimulation of the compound muscle action potential was evaluated on the right tibial/sciatic nerve.

### 2.2. Histopathology and Immunofluorescent Labeling

Biopsies were collected under general anesthesia from the left cranial tibial and gluteal muscles immediately following electrodiagnostic evaluation. Unfixed chilled and formalin fixed specimens were submitted by a courier service under refrigeration to the Comparative Neuromuscular Laboratory at the University of California San Diego. Upon receipt, the biopsy specimens were either flash frozen in isopentane precooled in liquid nitrogen then stored at −80 °C until further processed or paraffin embedded for routine processing. Cryosections were cut and stained by a standard panel of histochemical stains and reactions [11]. Additional cryosections were cut and incubated with monoclonal or polyclonal antibodies against laminin α2 (undiluted, 1B4, gift of Eva Engvall [12,13]), laminin γ1 (undiluted, 2E8, gift of Eva Engvall) and collagen VI (undiluted, 3G7, gift of Eva Engvall) from the affected dog and archived control muscle by immunofluorescent methods as previously described [14].

### 2.3. Animals and DNA Extraction

EDTA blood samples were collected from the affected dog, the mother of the affected dog, a paternal half-sister, the mother of the half-sister, and 10 additional unaffected and unrelated Italian Greyhounds. Genomic DNA was isolated from EDTA blood with the Maxwell RSC Whole Blood Kit using a Maxwell RSC instrument (Promega, Dübendorf, Switzerland).

### 2.4. Whole-Genome Sequencing

An Illumina TruSeq PCR-free DNA library with ~400 bp insert size of the affected dog was prepared. We collected 225 million 2 × 150 bp paired-end reads on a NovaSeq 6000 instrument (25.7× coverage). Mapping and alignment were performed as described [15]. The sequence data were deposited under the study accession PRJEB16012 and the sample accession SAMEA8157176 at the European Nucleotide Archive.

### 2.5. Variant Calling

Variant calling was performed using GATK HaplotypeCaller [16] in gVCF mode as described [15]. To predict the functional effects of the called variants, SnpEff [17] software together with NCBI annotation release 105 for the CanFam3.1 genome reference assembly was used. For variant filtering, we used 795 control genomes from wolves and dogs of diverse breeds (Appendix A).

### 2.6. Gene Analysis

We used the CanFam3.1 dog reference genome assembly and NCBI annotation release 105. Numbering within the canine *LAMA2* gene corresponds to the NCBI RefSeq accession numbers XM_022419950.1 (mRNA) and XP_022275658.1 (protein).

### 2.7. PCR and Sanger Sequencing

The *LAMA2*:c.3285G>A variant was genotyped by direct Sanger sequencing of PCR amplicons. A 304 bp PCR product was amplified from genomic DNA using AmpliTaq Gold 360 Mastermix (Thermo Fisher Scientific, Reinach, Switzerland) together with primers 5′-TGT GAC CCA AAG ACT GGT CA-3′ (Primer F) and 5′-AAA CAT GGT GCT TGC TTC ATC-3′ (Primer R). After treatment with exonuclease I and alkaline phosphatase, amplicons were sequenced on an ABI 3730 DNA Analyzer (Thermo Fisher Scientific). Sanger sequences were analyzed using the Sequencher 5.1 software (GeneCodes, Ann Arbor, MI, USA).

## 3. Results

### 3.1. Clinical History and Examination

A 4-month-old, male intact Italian Greyhound was presented for investigation of abnormal pelvic limb gait. The dog was reported to be hyporexic with occasional regurgitation, lethargic and exercise intolerant. The dog was obtained from an Italian Greyhound breeder at 2 months of age and noted to be smaller compared to its unaffected littermates. The abnormal short-strided gait was noticed shortly after adoption.

Physical examination revealed a poor body condition score (1/9) with generalized skeletal muscle atrophy. The dog would stand with mild kyphosis and valgus deformities (Figure 1). When ambulating the dog would maintain this posture, show a stiff pelvic limb gait with limited flexion of both stifles and hocks, and would externally rotate both tarsi during the early swing phase while externally rotating the stifles during the postural stance phase (Appendix A). The patellar reflexes were decreased bilaterally and the withdrawal reflexes were mildly decreased in the thoracic limbs. No other abnormalities were detected on physical, orthopedic, and neurological examinations.

### 3.2. Laboratory and Electrodiagnostic Examinations

Complete blood count revealed lymphopenia (1.96 × 10^9^/L, ref. range: 2.8–7.5 × 10^9^/L) and eosinopenia (0.16 × 10^9^/L, ref. range 0.8–2.0 × 10^9^/L). Serum biochemistry revealed mildly elevated alanine transaminase activity (ALT, 275 U/L, ref. range: 0–100 U/L) and markedly elevated CK activity (2960 U/L, ref. range: 10–200 U/L). Repeated serum biochemistry 3-months after initial samples revealed further increases in CK (6452 U/L) and ALT (855 U/L) activities. Serology for *Toxoplasma gondii* and *Neospora caninum* were negative.

Blood count, biochemistry, electrolytes, and CK activity from the mother of the affected dog, the paternal half-sister, and the mother of the half-sister were normal.

### 3.3. Electrodiagnostic Testing

Electromyography revealed fibrillation potentials and positive sharp waves in all axial and appendicular muscles tested on the right side. Motor nerve conduction velocity (MNCV) along the right tibial/sciatic nerve (i.e., stimulation sites, coxofemoral joint, stifle and tarsus; recording site, plantar interosseous muscle) was within or close to the reference range for the age of the dog proximally (46 m/s) but decreased distally (26.2 m/s). The evoked compound muscle action potentials (CMAP) were decreased in amplitude (hock, 1.5 mV; stifle, 2.4 mV; hip, 2.9 mV), but configuration was normal based on subjective comparisons with results obtained from normal animals [18,19]. Repetitive stimulation of the CMAP revealed a decremental amplitude of >10%. Motor nerve conduction velocity (MNCV) along the left ulnar nerve (i.e., stimulation sites: elbow joint and carsus; recording site: palmar interosseous muscle) was within or close to the reference range for a dog of this age proximally (40 m/s). The evoked CMAPs decreased in amplitude (elbow, 1.3 mV; carpus, 1.1 mV), but configuration was normal, based on subjective comparisons with results obtained from normal animals.

### 3.4. Clinical Outcome

At 8 months of age, the gait worsened with intermittent, bilateral pelvic limb lameness. A repeat orthopedic examination revealed grade 3 and grade 2 patellar luxation on the left and right pelvic limb, respectively. Body condition score remained poor at 2/9. The dog was still alive at the time of manuscript submission, at one year and six months of age. At that stage, he was occasionally stumbling in the thoracic limbs and was coping with two walks a day of approximately 1800 meters each. Occasionally, he was taken for approximately 4000 m walks on top of his regular walks, after which he seemed tired.

### 3.5. Histopathological Examination

Biopsies were evaluated from the cranial tibial and gluteus muscles with similar changes in both muscles. A marked variability in myofiber size was observed with numerous atrophic fibers (diameters < 10 µm) and scattered hypertrophic fibers (Figure 2). Mild endomysial fibrosis, scattered myofibers containing internal nuclei, and occasional necrotic fibers undergoing phagocytosis were observed. The pattern of changes were consistent with a congenital myopathy with a dystrophic phenotype.

### 3.6. Genetic Analysis

As clinical and histological findings resembled previously published cases of dogs with CMD [9,10,20], we hypothesized that the phenotype in the affected dog was due to a variant in one of the known CMD candidate genes [1]. We sequenced the genome of the affected dog and searched for private homozygous and heterozygous variants that were not present in the genome sequences of 786 control dogs and nine wolves (Table 1 and Appendix A).

This analysis identified a single homozygous private protein-changing variant in *LAMA2*. This variant, a nonsense variant, XM_022419950.1:c.3285G>A, is predicted to result in a premature stop codon, XP_022275658.1:p.(Trp1095*). The genomic designation of this variant is Chr1:67,883,271G>A (CanFam3.1). The presence of the variant in the affected dog was confirmed by Sanger sequencing (Figure 3A). A total of 14 available unaffected Italian Greyhounds including several close relatives of the affected dog were genotyped. 

The only dog with a heterozygous genotype for the *LAMA2* variant was the healthy dam of the case (Figure 3B). The other 13 unaffected dogs were homozygous for the wild-type allele.

### 3.7. Immunofluorescent Labeling of Laminin α2 in Muscle Cryosections

To obtain additional information on the functional impact of the *LAMA2* variant, we performed immunofluorescent labeling of muscle cryosections of the affected dog and an unaffected control. Using an antibody against laminin α2, staining of the muscle basal lamina was not detected in the affected dog but was prominent in the control (Figure 4).

## 4. Discussion

In this study, we describe a CMD in an Italian Greyhound dog associated with a *LAMA2* nonsense variant and laminin α2 deficiency. Descriptions of laminin α2 deficiency are rare in dogs with one recent description in a Staffordshire Terrier with a *LAMA2* gene deletion [10]. Another case was described in 2001 in a Brittany Spaniel–Springer Spaniel crossbred dog confirmed with immunofluorescence labeling for laminin α2 deficiency but for which the causative genetic variant was not yet identified [21]. *LAMA2*-related muscular dystrophies represent one of the most common forms of human CMD world-wide [22], so it seems possible that this form of CMD occurs more commonly in dogs but is not clinically recognized and underdiagnosed.

The *LAMA2* gene encodes laminin subunit α2, also referred to as merosin. Together with the laminin subunits β1 and γ1 (encoded by *LAMB1* and *LAMC1* respectively), laminin α2 forms the T-shaped heterotrimeric extracellular glycoprotein laminin-211 [23]. All 15 distinct known forms of laminins [24] serve two main purposes: They form a scaffold that connects the extracellular matrix to the cell surface and they act as attachment site for other proteins of the cellular matrix [25]. Laminin-211 serves this function in the basement membrane around the muscle fibers of skeletal muscles and in peripheral nerves [26]. It links to sarcolemmal α-dystroglycan and α7β1 integrin, building the dystrophin-glycoprotein complex, an essential element for sarcolemmal stability and muscle function [25,26].

Two distinct clinical phenotypes of *LAMA2*-related CMD were recognized in humans; a severe, early onset form, and a less severe later onset form [22,27,28,29,30,31,32,33]. The CK activity is persistently elevated in both forms. In the severe form, affected humans are symptomatic at birth with severe hypotonia and axial weakness. Joint contractures, respiratory insufficiency and scoliosis are common. Facial weakness, elongated face, macroglossia, protruding tongue and drooling are also typical. In the later onset form, presentation is during childhood and is related to delayed motor milestones. Scoliosis is not frequent and patients in this group are less likely to require ventilation. The milder form often presents with a phenotype suggestive of limb-girdle muscular dystrophy, with prominent joint contractures [30].

With the limited number of canine cases confirmed, it is hard to know the spectrum of clinical presentations in dogs with *LAMA2*-related CMD; however, so far, the canine cases described are clinically more consistent with the later onset mild human form. The complete absence of laminin α2 immunolabeling of muscle sections in affected human cases was described as indicative of more severe disease while partial labeling was associated with milder disease [22]. In both the dog of this report and in the previously published Staffordshire Terrier [10], a complete absence of laminin α2 was identified. Nonetheless, both canine cases were clinically closer to the mild form in humans. The lighter weight compared to that of humans and being quadruped may have contributed to achieving and maintaining ambulation and respiratory function in the affected dogs despite the complete absence of laminin α2.

In addition to skeletal muscle, laminin α2 is also expressed in Schwann cells of the peripheral nerve and astrocytes and pericytes of the capillaries in the brain [34]. Therefore, manifestations of central nervous system and peripheral nervous system involvement may be apparent including seizures [30,31] and sensorimotor demyelinating neuropathy [30,35]. The decreased distal nerve conduction velocity of the tibial/sciatic nerve and decreased CMAP in the dog of this report could be supportive of mild neuropathy; however, no nerve biopsies were obtained to confirm. In one reported cat with laminin α2 deficient CMD, a demyelinating polyneuropathy was confirmed in peripheral nerve biopsies in addition to the dystrophic myopathy [36].

Similar to human patients, CK activity in the affected dog remained persistently elevated. A persistently elevated CK activity in the clinical context of muscle weakness and atrophy should alert the clinician to the possibility of a neuromuscular disease. Patellar luxation was not observed at the initial presentation, but was identified 4 months later. Joint contractures are well recognized in both forms of the human disease, which may explain the development of patellar luxation and valgus deformity in this dog [22].

Whole genome sequencing identified a homozygous private nonsense variant in the *LAMA2* gene that is predicted to truncate 65% from the open reading frame of the wild type *LAMA2* transcript, XP_022275658.1:p.(Trp1095*). We assume that the premature stop codon results in a complete loss of function according to established guidelines for the interpretation of sequence variants used in human medicine [37].

## 5. Conclusions

The clinical and histopathological presentation, genetic findings and demonstrated absence of laminin α2 protein expression in skeletal muscle together with the existing knowledge of *LAMA2*-related CMD in dogs [10] and other species [22,38] establish *LAMA2*:c.3285G>A as causative variant for the observed CMD phenotype in the investigated dog. The identification of a causative variant enables genetic testing and the detection of heterozygous carriers, thus preventing the further unintentional breeding of affected dogs.

## Figures and Tables

**Figure 1 genes-12-01823-f001:**
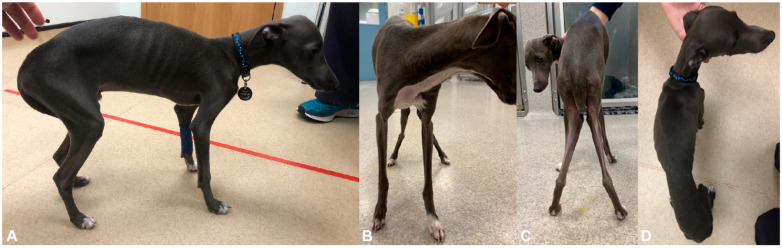
Clinical phenotype. Photos of affected dog illustrating kyphosis and generalized skeletal muscle atrophy on (**A**) trunk and lateral limbs, (**B**) thoracic limbs, (**C**) pelvic limbs, and (**D**) dorsum. Note genu valgum in panel (**C**).

**Figure 2 genes-12-01823-f002:**
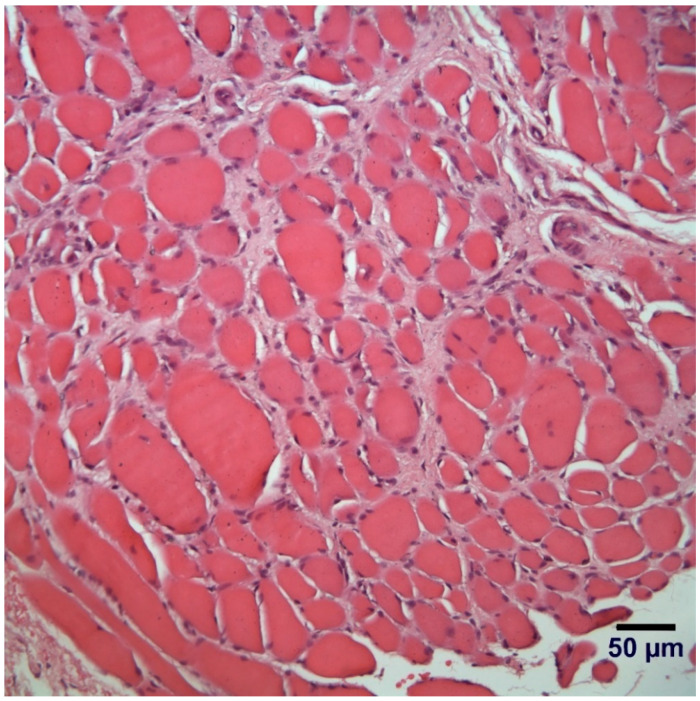
Hematoxylin and eosin (H&E) stained paraffin embedded section from gluteus muscle showing a marked variability in myofiber size, endomysial fibrosis and scattered myofiber showing internal nuclei. A form of congenital muscular dystrophy was suspected.

**Figure 3 genes-12-01823-f003:**
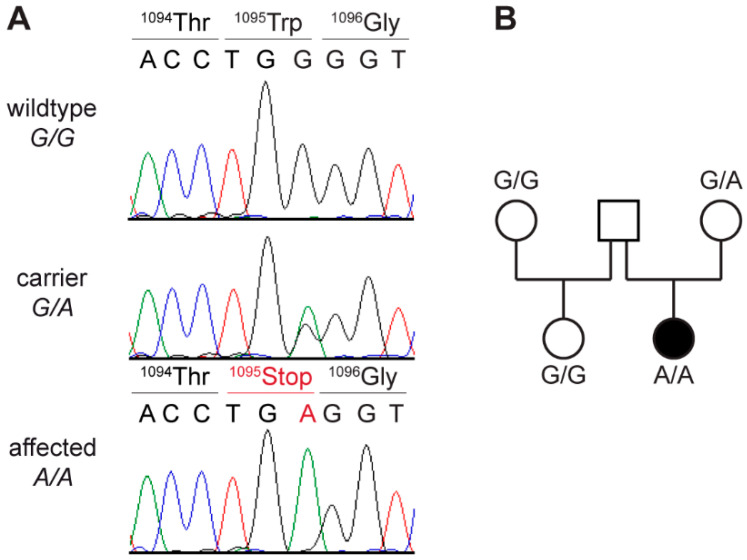
Details of the *LAMA2*:c.3285G>A variant. (**A**) Representative Sanger sequencing electropherograms of dogs with three different genotypes are shown. Amino acid translations are indicated. (**B**) Pedigree of affected Italian Greyhound. Genotypes at *LAMA2*:c.3285G>A variant are indicated. No sample of sire was available for genotyping. Dam was confirmed to be heterozygous, as expected for an obligate carrier of a monogenic autosomal recessive trait.

**Figure 4 genes-12-01823-f004:**
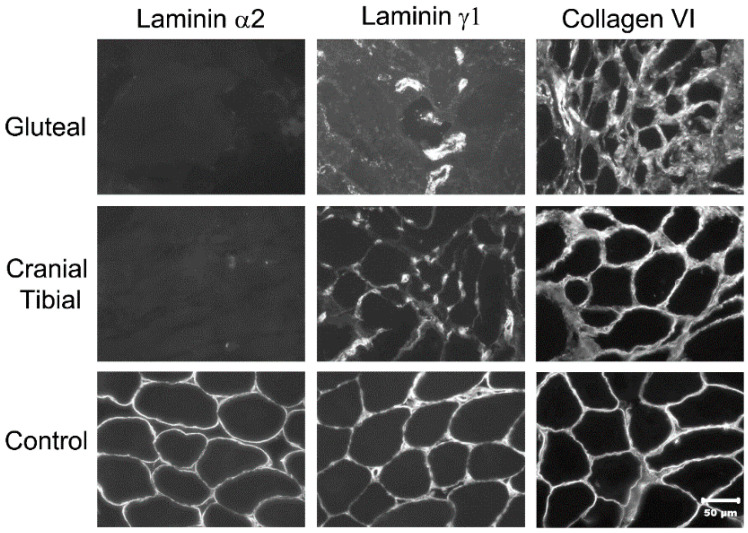
Immunofluorescent labeling of muscle cryosections from gluteal and cranial tibial muscles of affected dog and archived control muscle. Cryosections were incubated with antibodies against laminin α2, laminin γ1 and collagen VI. Labeling of basal laminina with antibody against laminin α2 was not detected in muscles from affected dog but was prominent in archived control muscle. Labeling with antibody against the ubiquitous laminin γ1 was reduced and labeling with antibody against collagen VI was similar to control. Reduced labeling of laminin γ1 most likely is a secondary consequence of laminin α2 deficiency as these molecules normally are parts of laminin-211 complex in basal lamina. Bar = 50 µm for all images.

**Table 1 genes-12-01823-t001:** Results of variant filtering in affected dog against 795 control genomes.

Filtering Step	Homozygous Variants	Heterozygous Variants
All variants in the affected dog	2,819,563	3,360,003
Private variants	1476	8957
Protein-changing private variants	6	68
in 39 known CMD candidate genes [1]	1	0

## Data Availability

The accessions for the sequence data reported in this study are listed in Appendix A.

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
