# Peer review of "LAMA2 Nonsense Variant in an Italian Greyhound with Congenital Muscular Dystrophy"

_genes, 2021, doi:10.3390/genes12111823_

Round 1

Reviewer 1 Report

General comments

This article describes a unique case of a young Italian Greyhound dog presented at the age of 4 months with postural and gait abnormalities, and generalized skeletal muscle atrophy. His parents were healthy.

The authors have carried out a thorough investigation by collecting clinical, paraclinical, histopathological and genetic data to eventually claim a robust and well-supported conclusion of a muscular dystrophy linked to the presence in its genome of a homozygous loss-of-function allele of the LAMA2 gene. This combination induced in this dog a primary deficiency of laminin 2 expression in skeletal muscle, reducing the expression of the laminin 211 complex.

The genetic characterization of the variant has been remarkably well performed, and the pipeline of variants filtering is nicely illustrated in Table 1, which emphasizes the power of an international whole-genome  sequence dataset that repeatedly proved its efficiency.

In light of the literature on the subject, the variant can be described as causal. The conclusions drawn from these data are therefore relevant and unambiguous. The manuscript is concisely written, cites the right references, is pleasant to read and uses original wording that demonstrates the authors' concern to make the reader understand each of the experimental steps cited. For example, it is common to read that a muscular dystrophy is accompanied by an increase in CK, without any reminder of the underlying mechanism, here summarized in lines 37 to 39. 

Here are a few points that may allow the authors to add or discuss some useful information.

1- Clinically, it would be very interesting to say something about the predicted impact of the mutation on cardiac function by analyzing an ECG or an echocardiographic recording. LAMA2 is highly expressed in the heart, with abnormalities infrequently described in the literature. This would be a great opportunity to do so.

2- Histopathologically, it would be interesting to document how this mutation impacts the dystrophin-glycoprotein membrane complex by a specific immunostaining.

3- In the conclusion, line 244, the authors describe this case as a form "clinically closer to the mild form in humans". I think this should be clarified. It seems that this dog is quite severely affected, probably with a level of severity between a dystrophic (DMD) and a myotubularin deficient  (MTM1) dog. It is likely that the clinical difference with the severe form in humans is due to the low body mass of this breed and the four-legged position of the animal, both of which protect the animal from ventilatory insufficiency caused by major dysfunction of the diaphragm, accessory ventilatory and axial muscles, the consequences of which are particularly deleterious in standing children.
In this dog, the involvement of all the paravertebral and facial muscles, including the tongue, the generalized muscular atrophy and the locomotor difficulties (also perhaps the cardiac dysfunction?) would be in favor of a marked myopathy that can probably be compared to the severe form in humans, whose early onset is shared.

4- It would be useful to know what has happened to this dog. Is he still alive, was he finally euthanized for medical reasons? If so, at what age and for what reasons? Did he reach puberty and could sperm be collected in order to be able to ultimately produce litters for putative preclinical trials of innovative therapies?

Minor points

Line 69: immunofluorescent

Lines 78-79: which epitope of the protein is target with 1B4 (which part of the protein? Before or after the created stop codon?)

Line 123: maintain

Line 160: with which results?

Line 164: any counts of atrophic fibers?

Line 187: I would write wild-type (adjective)

Lines 225, 226: sarcolemmal

Author Response

(1)

Clinically, it would be very interesting to say something about the predicted impact of the mutation on cardiac function by analyzing an ECG or an echocardiographic recording. LAMA2 is highly expressed in the heart, with abnormalities infrequently described in the literature. This would be a great opportunity to do so.

Response: Thank you very much for this comment. If the dog comes in for a follow-up examination, we will for sure consider to perform an ECG and a thorough examination of cardiac function.

(2)

Histopathologically, it would be interesting to document how this mutation impacts the dystrophin-glycoprotein membrane complex by a specific immunostaining

Response: We agree that it would be interesting to also look at the dystrophin-glycoprotein complex on the muscle sarcolemma. We focused primarily on extracellular matrix proteins as the quality of the biopsy samples was insufficient to evaluate sarcolemmal proteins such as dystrophin, sarcoglycans and dystroglycans. Please keep in mind that we only had “routine” samples for diagnostic purposes taken from the index case before we knew the causative genetic defect. If further cases will become available, we will for sure consider to perform the suggested additional investigations.

(3)

In the conclusion, line 244, the authors describe this case as a form "clinically closer to the mild form in humans". I think this should be clarified. It seems that this dog is quite severely affected, probably with a level of severity between a dystrophic (DMD) and a myotubularin deficient (MTM1) dog. It is likely that the clinical difference with the severe form in humans is due to the low body mass of this breed and the four-legged position of the animal, both of which protect the animal from ventilatory insufficiency caused by major dysfunction of the diaphragm, accessory ventilatory and axial muscles, the consequences of which are particularly deleterious in standing children.

In this dog, the involvement of all the paravertebral and facial muscles, including the tongue, the generalized muscular atrophy and the locomotor difficulties (also perhaps the cardiac dysfunction?) would be in favor of a marked myopathy that can probably be compared to the severe form in humans, whose early onset is shared.

Response: Thank you very much for this important comment with which we fully agree. We added a sentence to the discussion emphasizing that the lighter body weight and being quadruped are probably the main reasons for the slightly milder course of disease in the affected dogs compared to human patients.

(4)

It would be useful to know what has happened to this dog. Is he still alive, was he finally euthanized for medical reasons? If so, at what age and for what reasons? Did he reach puberty and could sperm be collected in order to be able to ultimately produce litters for putative preclinical trials of innovative therapies?

Response: The dog was still alive at 1 year and 6 months of age (line 159/160). Theoretically, sperm could be collected from this dog to start a colony. However, to the best of our knowledge this is very rarely done, at least in Europe. Nowadays, it might be easier to produce genome-edited dogs by CRISPR/Cas technology than starting a colony that would require breeding of at least 2 generations of dogs until homozygous mutant dogs can be expected.

(5)

Minor points

Line 69: immunofluorescent

Response: Revised accordingly.

(6)

Lines 78-79: which epitope of the protein is target with 1B4 (which part of the protein? Before or after the created stop codon?)

Response: The monoclonal antibody 1B4 was raised against protein extracts purified from human placenta. To the best of our knowledge, the exact epitope has never been precisely defined. We added the original reference from 1988 that describes the generation of this antibody.

(7)

Line 123: maintain

Response: Revised accordingly.

(8)

Line 160: with which results?

Response: We revised the short paragraph on clinical outcome and removed the statement on the recommended physiotherapy and hydrotherapy as we do not have any data on the outcome of these therapies.

(9)

Line 164: any counts of atrophic fibers?

Response: We did not attempt a formal morphometric analysis as this is rarely performed on routine diagnostic muscle biopsy specimens. The image in Figure 2 is representative of the entire field and a magnification bar is provided. The distribution of atrophic fibers and relative sizes can be estimated from this figure.

Line 187: I would write wild-type (adjective)

Response: Revised accordingly.

Lines 225, 226: sarcolemmal

Response: The typographical errors were revised accordingly.

Reviewer 2 Report

The manuscript describes the identification of a causal mutation in the LAMA2 gene causing muscular dystrophy in an Italian Greyhound. The research includes genetic analysis - sequencing of the entire NGS genome of a sick animal and comparing the obtained sequence with 795 genomes of dogs available in the database. On this basis, a mutation in the LAMA2 gene was identified causing an earlier appearance of the STOP codon in the transcript. The mother of the sick animal was also the carrier of this mutation. Additionally, immunofluorescence staining was performed, confirming the absence of Laminin 2 protein in the biopsy sections of the sick dog's muscle. The clinical picture as well as the biochemical analyzes of the blood of the sick animal indicate congenital muscular dystrophy. The presented results allow to confirm with a high degree of probability that it is the identified mutation that is responsible for the appearance of the disease in the tested dog. Identification of this mutation in a heterozygous form in the father of a sick animal would be even more certain, but the authors do not have material from the father, which should be considered a limitation of the study. The work is written in a clear and transparent manner and the presented studies form a logical sequence indicating the identified mutation as the cause of muscular dystrophy in the dog.

Minor comments:

line 95-99 Please describe how was the variants filtered (read depth, clusters)

line 101-103 Please provide the SNP coordinates (position on a chromosome)

Was the SNP deposited to database (for example European Variant Archive)?

Table S2 Please add the explanation to the table caption that analyzed mutation is indicated in yellow

Author Response

(1)

line 95-99 Please describe how was the variants filtered (read depth, clusters)

Response: A full description of the bioinformatics workflow was given in our earlier publication (ref. 15, Jagannathan et al. Anim. Genet 2019, 50:695-704). As the full description is rather long, we think it would be inappropriate to copy/paste long sections of texts from our earlier publication. This information is readily available from the cited reference.

(2)

line 101-103 Please provide the SNP coordinates (position on a chromosome)

Response: Chapter 2.6 (lines 101-103) in the methods section only specifies the used reference sequences. Variant coordinates are given in the results. We added the genomic coordinate to the results in lines 185/186. The genomic coordinate can also be retrieved from Table S2.

(3)

Was the SNP deposited to database (for example European Variant Archive)?

Response: Thank you for this comment. We fully agree that experimentally confirmed sequence variants should be deposited in a public database and rs numbers should be reported. However, unfortunately, the European Variant Archive (EVA) has a huge backlog in accessioning new dog variants. More than 2 years ago, we submitted >20 million variants from our 2019 DBVDC publication (Jagannathan et al. Anim. Genet 2019, 50:695-704) to EVA. These variants have still not been processed and have not received rs numbers. Due to this unfortunate situation at EVA, we currently do no longer submit new variants to EVA. We plan to do this immediately as soon as EVA will have accessioned our past variant submissions and established a working pipeline for submitting new variants.

(4)

Table S2 Please add the explanation to the table caption that analyzed mutation is indicated in yellow.

Response: Revised accordingly.